Research

# TElehealth in CHronic disease: mixed-methods study to develop the TECH conceptual model for intervention design and evaluation

Chris Salisbury,[1] Clare Thomas,[1] Alicia O'Cathain,[2] Anne Rogers,[3] Catherine Pope,[3] Lucy Yardley,[4] Sandra Hollinghurst,[1] Tom Fahey,[5] Glyn Lewis,[6] Shirley Large,[7] Louisa Edwards,[1] Alison Rowsell,[4] Julia Segar,[8] Simon Brownsell,[2] Alan A Montgomery[9]

▶ Prepublication history and additional material is available. To view please visit the journal (http://dx.doi.org/10.1136/bmjopen-2014-006448).

For numbered affiliations see end of article.

**Correspondence to**
Professor Chris Salisbury;
c.salisbury@bristol.ac.uk

## ABSTRACT

**Objective:** To develop a conceptual model for effective use of telehealth in the management of chronic health conditions, and to use this to develop and evaluate an intervention for people with two exemplar conditions: raised cardiovascular disease risk and depression.

**Design:** The model was based on several strands of evidence: a metareview and realist synthesis of quantitative and qualitative evidence on telehealth for chronic conditions; a qualitative study of patients' and health professionals' experience of telehealth; a quantitative survey of patients' interest in using telehealth; and review of existing models of chronic condition management and evidence-based treatment guidelines. Based on these evidence strands, a model was developed and then refined at a stakeholder workshop. Then a telehealth intervention ('Healthlines') was designed by incorporating strategies to address each of the model components. The model also provided a framework for evaluation of this intervention within parallel randomised controlled trials in the two exemplar conditions, and the accompanying process evaluations and economic evaluations.

**Setting:** Primary care.

**Results:** The TElehealth in CHronic Disease (TECH) model proposes that attention to four components will offer interventions the best chance of success:
(1) engagement of patients and health professionals, (2) effective chronic disease management (including subcomponents of self-management, optimisation of treatment, care coordination), (3) partnership between providers and (4) patient, social and health system context. Key intended outcomes are improved health, access to care, patient experience and cost-effective care.

**Conclusions:** A conceptual model has been developed based on multiple sources of evidence which articulates how telehealth may best provide benefits for patients with chronic health conditions. It can be used to structure the design and evaluation of telehealth programmes which aim to be acceptable to patients and providers, and cost-effective.

## Strengths and limitations of this study

- This paper describes the development and use of an evidence-based conceptual model for the effective use of telehealth among patients with chronic conditions.
- Having a conceptual model provides a framework for intervention development and evaluation.
- The model is now being evaluated through parallel randomised controlled trials in two exemplar chronic conditions.
- In order to develop a model which is clear, simple and generalisable, there is a risk of over-simplification of the multiple mechanisms by which telehealth may have its effects.
- The strength of evidence available to justify different components of the conceptual model is variable.

## INTRODUCTION

### The role of telehealth in chronic health conditions

There is international interest in the potential of telehealth to support the management of patients with chronic health conditions. Telehealth refers to the use of electronic and telecommunication technologies to support healthcare at a distance from the patient. This reflects a recognition that, as the population ages, the needs of the increasing number of people with chronic conditions are likely to overwhelm the capacity of conventional healthcare services designed around scheduled one-to-one and face-to-face appointments between patients and doctors. In the UK, 30% of the population have at least one chronic condition and they account for 70% of total health services expenditure.[1] There is a need to harness the potential of technology to support people to manage themselves in their own homes. This has potential to shift

the locus of control so that, through better access to information, people can become experts in their own care. Provision of healthcare at a distance (eg, through telemonitoring) could in theory be more accessible, efficient and responsive than patients or professionals having to travel for face-to-face appointments.

Considerable resources have been committed to implementing different forms of telehealth for chronic conditions. For example, in the USA, the Veterans Health Administration introduced a national home telehealth programme which had enrolled about 50 000 patients by 2011;[2 3] the Renewing Health Consortium is developing and testing telehealth programme in nine European countries,[4] while in the UK, the Whole System Demonstrator project was established to provide telehealth at scale for patients with conditions such as heart failure or chronic lung disease.[5–7]

## Evidence of benefits

Although the potential benefits of telehealth in chronic condition management have been rehearsed for at least 20 years, evidence to support these arguments is limited.[8–10] Systematic reviews have been conducted for specific chronic conditions, along with overviews which have combined findings from a range of conditions; these have concluded that the evidence in favour of telehealth is weak and inconsistent.[8 9 11–16] Evidence of effectiveness is stronger for some conditions (eg, heart failure) than it is for others (eg, diabetes). Some studies report positive findings while others do not and it has been difficult to identify a pattern in terms of disease, type of technology or patient characteristics to explain these inconsistencies. There is a lack of evidence about mechanisms of action and about wider impacts of telehealth on utilisation of other healthcare services.[9] There is inconsistent reporting of outcomes, suggesting a lack of clarity about the intended benefits of telehealth and making it difficult to compare studies. Evidence about cost-effectiveness or of successful wide scale implementation is particularly limited.

## The need for a conceptual model

Telehealth is a complex intervention[10 17] involving a number of interacting components, such as the type of technology, the infrastructure, the human support available and the capabilities of the patient in relation to the technology. For any individual, telehealth is likely to be only one facet of the healthcare they receive, so telehealth cannot be understood in isolation from the healthcare system in which it is provided.

Over the past 15 years, there has been increasing awareness of the importance of theory both in the development and evaluation of complex interventions.[18] Theory is needed in order to understand the relationship between context, mechanism of action and intended outcomes, but this has largely been neglected in the field of telehealth.[19–21] While there are well-recognised theories in related topics such as

behavioural change (eg, the Theory of Planned Behavior,[22] the Behaviour Change Wheel,[23] Ritterband[24]), and why technologies get used (eg, the Technology Acceptance Model[25]), there is no overarching theory which connects these and other elements (such as coordination between service providers) essential to chronic disease management in the context of telehealth.

What is needed is a clear conceptual model for how and why a telehealth intervention for patients with chronic conditions is intended to have specified beneficial effects. Making explicit the theoretical chain of causation by which an intervention is intended to lead to its effects focuses attention on the most important features of the intervention that need to be delivered for it to be effective. A conceptual model also provides a framework for evaluation by identifying the contextual factors, steps in the causal chain and most important, outcomes that need to be assessed. To be practically useful, a conceptual model should be sufficiently generalisable to apply to a range of conditions, types of interventions and healthcare settings.

This paper describes the development of a conceptual model for the role of telehealth in the management of chronic conditions. This was developed to inform the design of an intervention to support people with two exemplar conditions: raised cardiovascular disease risk (due to risk factors such as hypertension, smoking, obesity and hyperlipidaemia) or depression. These exemplars were chosen to represent very different types of conditions which would test the generalisability of the model; however, both conditions are common and in both conditions there was existing evidence that some form of telehealth could be effective.[26 27] By taking into account the views of the patients and providers, and considerations about cost as well as evidence of effectiveness, the intention was to develop a model for interventions which are likely to be suitable for implementation on a wide scale, acceptable to stakeholders and cost-effective.

## METHODS
### Evidence review

The model was based on several sources of evidence. The methods and results for each strand of evidence are summarised below, but are described in more detail elsewhere.

1. A *meta-review and realist synthesis* of existing quantitative and qualitative evidence on telehealth for chronic conditions[16]—this consisted of an overview of existing systematic reviews of telehealth interventions. We focused on reviews of chronic conditions generally rather than in relation to specific conditions. We included telephone and internet-based interventions (such as telecoaching, telephone/internet counselling and follow-up) and telemonitoring of symptoms and vital signs, but not telemedicine

approaches where technologies are used to share information between healthcare providers. We searched MEDLINE, CINAHL, EMBASE, AMED, PsycINFO and the Cochrane Library databases for high-quality systematic reviews about telehealth and chronic conditions published in English, between January 2005 and March 2010. Two reviewers independently reviewed abstracts and (where relevant) full papers and extracted data onto a standardised form. We supplemented the metareview with a new systematic review to look in more detail at studies of telehealth interventions focused on telehealth interventions for prevention of cardiovascular disease.[28] In addition, we identified and reviewed published qualitative studies of patients' experience of using telehealth interventions. In total, we included 16 systematic reviews (representing 662 quantitative studies) and 29 qualitative studies. We combined these sources of data in a realist synthesis in which we sought to identify mechanisms of action of telehealth in chronic conditions. Realist synthesis is an approach reviewing research evidence on complex interventions in order to provide an explanatory analysis for how and why they work (or do not work) in particular contexts or settings.[29]

2. A *qualitative study* of the potential role of telehealth in chronic conditions[30]—this involved interviews and observation with patients as well as doctors and nurses providing primary care for patients with chronic conditions, and health information advisors who provided an existing telephone-based health coaching and care management service for patients with chronic conditions, such as heart failure or diabetes.[31] Semistructured interviews were conducted with 38 patients and 68 health professionals, and observations were undertaken at a centre providing telehealth. The research took place between April 2010 and March 2011. Thematic analysis of qualitative data was undertaken.

3. A *survey of patients* to assess relationships between patient characteristics, health needs, difficulties with access to healthcare, attitudes towards and availability of various technologies, and interest in using different types of telehealth.[32] Patients with either raised cardiovascular risk (n=872) or depression (n=606) were identified and randomly sampled from 34 general practices in two areas of the UK and invited to complete a questionnaire.

4. Comparison with other models of chronic disease management—in order to take account of and compare our emerging conceptual model with existing models and frameworks, we familiarised ourselves with other widely used models of chronic condition management, particularly (but not exclusively) those relating to the use of telehealth. We wanted to identify common factors in these models which appeared to be associated with improved care and benefits for patients.

5. *Analysis of national guidelines*: in order to apply the model to our exemplar conditions, we identified the main recommendations and priorities for treatment from the current UK guidelines and compared these with guidelines from the USA and Europe. We cross-referenced these recommendations with our metareview to identify evidence for the effectiveness of telehealth interventions (eg, the use of online programmes to deliver cognitive behavioural therapy for depression; the use of home monitoring of blood pressure in patients with hypertension).

## Synthesis

We synthesised the findings from our evidence review in two stages. First, it was clear from the metareview and the qualitative study that engagement from both patients and professionals appeared to be key to the success of a telehealth intervention. We, therefore, used a modified PRECEDE-PROCEED[33] approach to intervention development in which we used the insights from our evidence sources to map the predisposing, enabling and reinforcing factors that determine engagement with telehealth, creating separate 'maps' for patients and health professionals. Predisposing factors provide the motivation to act in some way, enabling factors are those that make it possible to carry out the action and reinforcing factors influence the likelihood that one will perform the behaviour in the future based on positive or negative feedback. Through discussion within the research team, we listed and grouped themes from the literature reviews, qualitative research and patient surveys, cross-referenced to the sources of evidence. Next, commonalities across these three sources of evidence were highlighted and key themes relating to engagement with telehealth were identified. These key themes were then independently organised into predisposing, enabling and reinforcing factors by members of the research team familiar with the PRECEDE-PROCEED[33] definitions. Since it is possible that the same information can first serve as a predisposing factor and then later as a reinforcing factor, differences in classification, although rare, were resolved through discussion. Nonetheless, the real importance of classifying information into these types of causal factors was to devise temporally appropriate strategies to enhance motivators of and mitigate barriers to the target behaviour.

Second, we developed a draft model for the use of telehealth to support the management of chronic conditions which encapsulated the main findings from the evidence review. We discussed the findings from the various studies within the research team to describe the hypothesised relationships between different constructs in a schematic manner. Several different layouts and versions of the model were discussed iteratively in meetings within the research team as we critiqued and sought to improve the model. Finally, we convened an intensive 1 day workshop for a wide range of stakeholders (n=38) including patients, care providers, managers, commissioners of services, independent academics and the research team. We presented the

findings of the evidence review and the draft model to the stakeholders, who then discussed it in small groups and provided feedback. We used this to refine the final model, which we labelled the TECH model (TElehealth in CHronic Disease).

### Using the model to design an intervention

The research team used the TECH conceptual model to design a telehealth intervention known as the Healthlines Service. This was designed to be delivered by NHS Direct, which (at the time the intervention was designed) provided health information and advice throughout England based on a network of telephone call centres and an associated website. The intention was to design an intervention that would be likely to be cost-effective by maximising patient benefit at minimum cost and which could feasibly be rolled out quickly on a national scale if it proved to be effective. For these reasons, the design of the intervention sought to incorporate technologies which were already available and approaches for which there was already some evidence of effectiveness. We avoided cutting-edge technologies that were not already developed or tested, and high-cost solutions that would be unlikely to be widely available or deliverable to large numbers of patients. In order to maximise population benefit, the aim was to focus on the large number of patients at moderate risk of health problems (eg, patients with hypertension and other cardiovascular risk factors) rather than the small number of patients at high risk (eg, patients who have already had a stroke).

The research team used the patient and health professional 'maps' generated through the PRECEDE-PROCEED method to develop strategies to promote engagement with the telehealth interventions by addressing each of the predisposing, enabling and reinforcing factors previously identified.

### The model as a framework for evaluation

The TECH conceptual model was used to provide a framework for evaluation by describing the extent to which each element of the model was successfully delivered and the intended outcomes that were achieved. The Healthlines Service is being evaluated within two pragmatic parallel randomised controlled trials and the accompanying processes and economic evaluations. We recruited 43 general practices providing primary healthcare in three areas of England. Adult patients from these practices with either (A) raised risk of a first cardiovascular event (10-year risk >20%) or (B) depression were recruited to take part and were individually randomised to receive either usual primary care plus extra support from the Healthlines Service or usual primary care alone. The protocol for these trials has been published (Trial Registration: Current Controlled Trials: cardiovascular disease risk trial ISRCTN27508731 and Depression trial ISRCTN14172341).[34]

## RESULTS

### Evidence review

*Metareview, realist synthesis, qualitative study and quantitative patient survey:*

Key findings from these studies are summarised in box 1.

### Existing models of chronic condition management

We identified a number of existing models for the management of chronic conditions, but the dominant approach is the Chronic Care Model (CCM).[35] A number of studies have suggested that programmes based on the CCM can improve health outcomes for a range of chronic conditions, although it is uncertain which components of the model are most important or whether all are necessary.[36–38] The CCM includes elements which relate to national aspects of the healthcare system and does not in itself provide a model for the design of telehealth interventions. Between 2003 and 2007, the Veterans Administration introduced a national home telehealth programme, Care Coordination/Home Telehealth (CCHT),[2] which was strongly influenced by the CCM but applied the concepts more specifically to telehealth applications in a US context.

### Review of national guidelines

In order to apply a conceptual model to a specific condition, the key health problems and care needs must be identified. For raised cardiovascular disease risk, international guidelines suggested that these were the modifiable risk factors of hypertension, smoking, obesity, raised cholesterol and lack of exercise.[39–43] Evidence-based priorities for intervention included optimising drug treatment in order to achieve blood pressure targets; ensuring medication adherence; providing nicotine replacement therapy for smokers along with behavioural support; providing advice about diet and exercise, and referral to weight management programmes for obesity; and ensuring that statins were prescribed and taken.

For depression, the priorities for intervention included offering psychological therapies, such as cognitive behavioural therapy and/or antidepressant drug treatment with intensity of treatment tailored in relation to need; having relapse prevention strategies; ensuring medication adherence; offering peer support; avoiding alcohol misuse; encouraging exercise and assessing suicidal risk.[44 45]

### Synthesis and developing the model

Figure 1 shows the final TECH model illustrating the key components and the relationships between them, which we hypothesise will deliver cost-effective improvements in chronic disease management using telehealth. In summary, this model proposes that interventions to promote self-management, optimisation of treatment and care coordination are all essential aspects of chronic disease management, which are likely to lead to improved health outcomes, patient experience, access to

**Box 1**  Key findings from the metareview, qualitative study and patient survey

Metareview[16 28]

► Some evidence of improvements in clinical outcomes.
► Much of the primary research is of poor quality and limited to short-term effects.
► Evidence about impact on the wider healthcare system and cost-effectiveness is sparse.
► Inconsistent findings about effectiveness and resource utilisation, with few clear patterns in terms of types of patient, disease or technology associated with benefits.
► Many telehealth interventions for chronic conditions have struggled to engage both patients and healthcare professionals, with low uptake and high dropout rates.
► Simple technologies, especially those based on telephone support, have at least as strong an evidence base as more sophisticated technologies such as telemonitoring.
► Telephone support seems to enhance the benefit of web-based technology.

Realist synthesis

This suggested three key mechanisms by which telehealth worked to improve health outcomes:

► Relationships: good connections between patients, peer groups and/or professionals provide support.
► Fit: acceptability, ease of use and integration into everyday routines were important to both patients and professionals.
► Visibility: monitoring provides feedback, reinforcement and prompts to change behaviour but can also have negative connotations of surveillance.

Qualitative study[30]

► Nurses and doctors working in primary care were ambivalent about the contribution of telehealth to chronic condition management, because of concerns about the lack of evidence of benefit, duplication of their own work and a threat to their role.
► There is a need to take account of how new telehealth programmes integrate with existing health system structures.
► Patients were more likely to trust a telehealth system if it is endorsed by their usual primary care providers.
► Patients valued a personal approach based in human interaction.

Patient survey[32]

► There was moderately strong interest in telehealth support for chronic conditions across all age groups.
► There was greatest interest in telephone and internet-based interventions, and minimal interest in social media, particularly amongst older patients with chronic conditions.
► There was little relationship between healthcare need or difficulties in accessing healthcare and interest in telehealth.
► The most important constructs associated with interest in telehealth were confidence in using the technology and perceived advantages and disadvantages of telehealth.
► Interest in telehealth was not related to patient sociodemographic variables, after adjusting for modifiable factors such as access to and confidence in using the technology.

care and more cost-effective delivery of care. These benefits are more likely to be achieved if the service is delivered in an integrated way with other healthcare providers and the effectiveness of telehealth is likely to be moderated by the extent of patient and provider engagement, and also moderated by characteristics of patients and the healthcare system.

These components are described in more detail below.

## Engagement of patients and primary care providers

The literature metareview highlighted that many telehealth interventions have been unsuccessful because of low uptake by patients and high rates of dropout. Both our qualitative research and the patient survey illustrated the range of factors that act as motivators or barriers to patients using telehealth. These are summarised in box 2 based on our PRECEDE-PROCEED map of predisposing, enabling and reinforcing factors for patients.

With regard to healthcare professionals, our qualitative research indicated that many were unenthusiastic and in some cases, resistant towards telehealth interventions. Our PRECEDE-PROCEED map for professionals identified several factors that were likely to influence engagement in telehealth. These included the belief that medicine should be evidence-based and scepticism about the evidence for telehealth (predisposing factor), concerns about duplication of care (predisposing), the need for technology to be simple and reliable (enabling), and the importance of clarity of roles for conventional and telehealth providers, and good communication between them (reinforcing).

## Effective chronic disease management

Our evidence synthesis and review of existing models of chronic condition management suggested that strategies that contribute to effective care and which could be delivered via telehealth can be summarised under three headings: promoting self-management, optimising treatment and care coordination. The various strategies that comprise each of these headings are shown in box 3, along with citations for specific studies or reviews that provide evidence of effectiveness for each element (not necessarily in the field of telehealth).

## Partnership

Our qualitative research highlighted that a telehealth intervention is just one aspect of the healthcare provided to a patient with a chronic condition. These patients are likely to continue to get the majority of their care from their family practitioner, with whom they may have had a long-term relationship and whom they will continue to consult for reasons apart from their chronic condition. In addition, many patients with chronic conditions are likely to be receiving help from hospital specialists, and other healthcare and social care agencies.

However, our evidence review suggested that many previous telehealth interventions appear to have failed because they were designed in isolation from the rest of the healthcare system, leading to duplication of effort, lack of coordination between providers, inefficiency and confusion for patients. This is likely to reinforce the

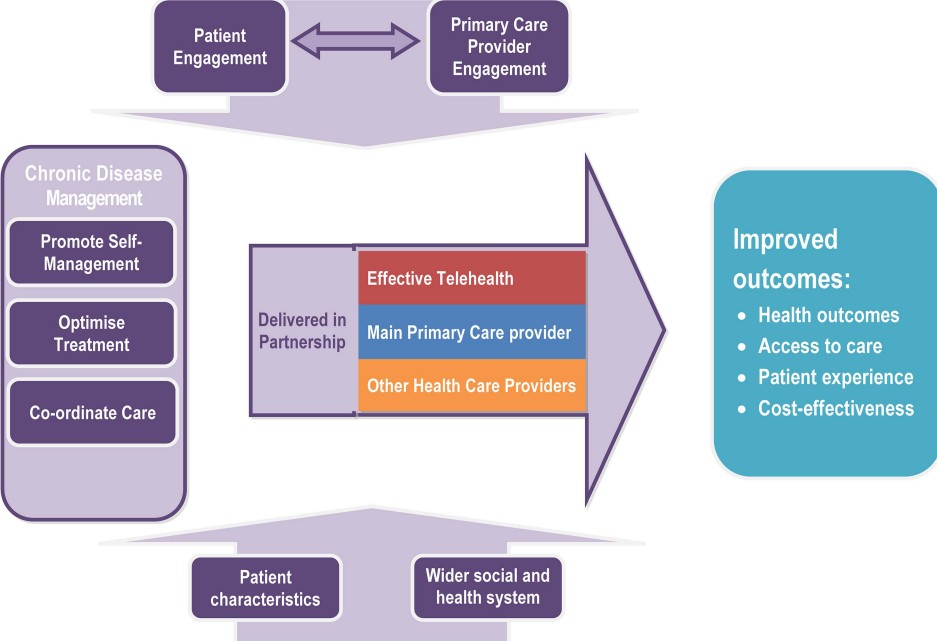

**Figure 1** The TElehealth in CHronic disease (TECH) model for telehealth to support patients with chronic conditions.

resistance expressed by other healthcare providers. Our qualitative research showed that these other providers may perceive the telehealth intervention to be an unnecessary interference in their area of responsibility, possibly representing a threat to their future role.

Therefore, it is important for a model for telehealth interventions to emphasise that telehealth should be delivered in partnership, identifying the role that telehealth can play to support rather than compete with patients' main primary healthcare providers.

## Context: characteristics of patients and wider social and health system

The patient survey and the literature review both indicated that characteristics of patients are likely to have an

impact on how telehealth affects outcomes. These include sociodemographic characteristics, particularly age, the nature of their chronic condition and the severity of their condition. The design of a telehealth intervention must also take account of the wider social and health system context.[62 63] For example, a programme

---

**Box 2** Predisposing, enabling and reinforcing factors to the use of telehealth by patients

**Predisposing**
► Attraction of having support for health problems on demand, having more time, getting greater support.
► Patients having a clear understanding of why they have been offered telehealth treatment.
► Confidence in ability to use the technology.
► Being reassured about privacy and confidentiality.

**Enabling**
► Good access to fast reliable internet connection.
► Technology which is simple and inexpensive, not complicated to use.

**Reinforcing**
► Benefits of having regular review.
► Importance of self-monitoring which promotes continued engagement.
► Encouraging patient activation and involvement rather than passive reminders.

---

**Box 3** Components of effective chronic condition management

**Promoting self-management**
► Behaviour change techniques, for example, stimulus control, problem solving, cognitive restructuring, goal setting.[46 47]
► Self-monitoring.[27 48 49]
► Provide patient information.[50 51]
► Promote self-efficacy.[52–54]
► Shared decision-making.[51]
► Motivational interviewing.[46 47]
► Personal support from health professionals.[55 56]

**Treatment optimisation**
► Risk stratification with case management for complex patients.[39 57]
► Treatment intensification.[39 44 56 58]
► Use of evidence-based guidelines and protocols.[44 56]
► Regular review.[39 51 58]
► Promote medication adherence.[47 51]
► Share treatment recommendations with patients.[59]

**Care coordination**
► Interventions that included multiple reinforcing components.[47 51 55]
► Shared records, information and treatment recommendations between patients, primary care and the telehealth provider.[2 54]
► Communication (remote and face-to-face) between the telehealth provider and primary care.[2]
► Regular monitoring of system performance.[38 60]
► Seek to support rather than duplicate primary care.[61]

designed to work within a health system context with a strong primary care foundation may need different features from one designed for a system in which patients consult different hospital specialists for each of their chronic conditions. Similarly, a system which assumes that patients have access to fast and reliable internet connections will not work where this does not apply. Finally, different funding models for healthcare create different financial incentives for providers and patients, which may have a major influence over how telehealth systems are implemented.

## Specifying outcomes

The TECH model depicted in figure 1 seeks to capture the four components of the model in a way that is conceptually clear, simple and generalisable. It also proposes the improved outcomes that telehealth interventions are intended to deliver for patients with chronic conditions. These are improved health outcomes, access to care and patient experience, and care provided in a way which is cost-effective. One criticism of earlier research on telehealth interventions has been the lack of consistency in reporting outcomes[8] and this model provides a framework for the outcomes that should be assessed in future evaluations, as well as potential mediators in order to gain understanding of the mechanism of action.

## Using the model to develop a telehealth intervention

We used the conceptual model to develop telehealth intervention programmes to support the management of patients with (A) raised cardiovascular risk or (B) depression. We used the same model to design interventions which were similar in concept but different in terms of detailed content to address each of the priority health and care needs for these two groups of patients based on our review of national guidelines.

Table 1 provides examples of how we devised strategies to be delivered within the Healthlines Service to populate the conceptual model for the intervention to be used for cardiovascular risk. Online supplementary appendix 1 provides an expanded and more comprehensive list of the strategies we used for both raised cardiovascular risk and depression; the Healthlines Service has also been described in detail elsewhere.[34]

## Use of the TECH model for evaluation

The TECH model proposes four main outcomes resulting from telehealth interventions for chronic disease, the first of which is improved health outcomes. For the cardiovascular trial, the primary outcome is cardiovascular risk status 12 months following randomisation. For depression, the primary outcome is a clinically significant improvement in depression. Secondary outcomes for both trials include health-related quality of life, measures of access to healthcare and patient satisfaction with healthcare. An economic analysis will assess cost-effectiveness over the 12 months of the trial. In the cardiovascular risk trial we will also model the long-term costs and benefits of the intervention after taking into account the predicted number of strokes and heart attacks over the next 10 years.[34]

Alongside the randomised controlled trial, a process evaluation will explore the extent to which the intervention was delivered as intended and whether it led to the expected changes at each step of causal chain hypothesised by the conceptual model. It assesses patient characteristics and health service context, patient and primary care engagement, patient self-management, treatment optimisation, care coordination and partnership with other healthcare providers, as well as the primary and secondary outcomes described above. These are assessed using validated measures, where possible. Qualitative research through interviews with patients, primary care health professionals and Healthlines advisors are conducted to understand in greater detail how the service was delivered, barriers and facilitators to implementation, and how and why the intervention did or did not appear to be effective from the perspectives of those delivering and receiving it.

## DISCUSSION

### Principal findings

This article describes the development of the TECH conceptual model for the effective use of telehealth among patients with chronic conditions and illustrates how it has been used to develop telehealth interventions for patients at either raised risk of cardiovascular disease or depression, and also to design the evaluation of those interventions. If these evaluations for different chronic conditions are positive, this will provide support for the model about how this type of telehealth intervention works, suggesting it can then be applied to other chronic conditions.

Alternatively, if the intervention is unsuccessful, it will be possible to assess each of the processes in the hypothesised causal chain in order to determine whether the intervention was not delivered as intended or whether the assumed causal relationships were incorrect. For example, the model posits that one way in which telehealth works is by allowing people to monitor their own health, which will lead to changes in their behaviour and this will have a positive impact on their health. Having a model highlights the need to assess the extent to which participants actually did self-monitoring as intended, whether this was associated with behavioural change and whether this led to improved health outcomes. This kind of approach provides a framework for correction and adaptation of an intervention through understanding which intervention components are more or less effective at impacting proximal outcomes in the causal chain.[65]

### Strengths and limitations

The strength of this research is that we have used diverse sources of evidence to develop a conceptual

**Table 1** Use of the TECH model to design the Healthlines telehealth intervention for patients with raised cardiovascular risk

| Model element | Strategies included in intervention |
| --- | --- |
| **Engagement** | |
| Patient | Healthlines advisors provide simple welcome pack and technical support to overcome lack of confidence in technology |
| | Encourage sense of personal care through seeking to maximise continuity of care from one named Healthlines advisor |
| Health professional | All communications seek to reinforce message that the Healthlines Service is supporting and delivered alongside primary care |
| | Messages to primary care emphasise evidence-based nature of interventions and guidance |
| **Promoting self-management** | |
| Behaviour change techniques | Healthlines cardiovascular intervention adapted from the Duke self-management package,[64] which uses scripts for advisors based on psychological principles of behaviour change. Intervention is tailored to patient's needs and goals |
| Self-monitoring and feedback | Provide patients with free BP monitors and website to log readings which gives immediate feedback and graphical display about whether BP is above or below target (see online supplementary appendices 2 and 3) |
| Provide patient information | Healthlines advisor works with patients to identify goals and then emails them links to further resources available on the internet, which have been quality assessed (eg, diet advice, risk calculators, videos, patient forums) |
| **Treatment optimisation** | |
| Risk stratification | Calculate cardiovascular risk. Level of intervention guided by level of risk factor with escalation to GP for patients at high risk |
| Treatment intensification | Monthly review of BP using online log of BP readings, protocol driven advice to GP to intensify treatment each month if targets not met |
| Promote medication adherence | Monthly review of medication adherence, scripts use evidence-based strategies to improve adherence, GPs advised by email if patients appeared to be non-adherent |
| **Care coordination** | |
| Shared records | All treatment recommendations shared with both primary care provider and patient. A summary of recent BP records from patient web portal is sent to GP when treatment change is recommended |
| Regular monitoring of system performance | Reporting module which allows monitoring of management programme (eg, of number of patients who have been telephoned, number actively self-monitoring BP) |
| **Partnership** | |
| | All communications are shared between Healthlines, GP and patient. Communication is two way: GPs can contact Healthlines, for example, to change a patient's BP target |
| | GPs and service managers involved in designing the Healthlines intervention |
| **Context** | |
| | Not all patients in UK have access to reliable internet connections. It is important to describe the characteristics of patients who take part, for evaluation |

BP, blood pressure; GP, general practitioner.

model which creates a framework for intervention development and evaluation. Each of the components of the model can be justified from our own research and evidence from previous literature.

Although it is arguable that the TECH model could be applicable not only to telehealth but to all chronic disease management programmes, the model draws attention to topics which are particularly important for telehealth (such as the need for partnership with primary care providers and attention to patient engagement) but which have been neglected in many previous telehealth interventions.

Recognising that the simplest models have the greatest utility, we sought to provide a simple graphical depiction of the hypothesised causal chain in a successful telehealth intervention. However, we recognise that the model diagram oversimplifies the multiple potential mechanisms by which a telehealth intervention may have its effect. There are likely to be associations and interactions between different elements of the model, and both recognised and unrecognised confounding factors. However, to indicate all of these potential relationships in the model would, in our view, reduce its usefulness in providing a framework.

A further limitation is that the strength of underlying evidence to support each of the components of the model is variable. For example, evidence of the benefit of patient self-monitoring is strong for some chronic conditions, but not all, and although providing patient information and shared decision-making are viewed as

important aspects of chronic condition management in the CCM and other similar models, the evidence that these strategies lead to improved patient outcomes is limited. Nevertheless, we have sought to include components in the model where the overall weight of evidence supports their value.

### Relationship to previous studies

There are several existing models of behaviour change based on psychological theory which have been applied to or are relevant to telehealth.[22–24] However, behaviour change is only one aspect of the TECH model and this is not its main purpose. The TECH model is intended to provide a framework for the design and evaluation of telehealth services at scale within healthcare systems, taking into account a much wider range of factors such as the potential efficiencies gained through better coordination of services.

Several previous authors have argued for the importance of theory in designing telehealth interventions,[19 60 66] and there are also existing frameworks for the assessment (rather than the design) of telehealth for chronic conditions, such as the Model for Assessment of Telemedicine (MAST).[67] The intervention which is most relevant to our study and well described in terms of its underlying theoretical basis is the Comprehensive Health Enhancement Support System (CHESS), an umbrella term for several e-health programmes combining information, adherence strategies, decision-making tools and support services.[65 68] Like the Healthlines intervention described here, CHESS was developed by combining several intervention features, each of which had some theoretical justification. However, CHESS was developed without any clear theory about how the programme features related to each other[65] and the TECH model underpinning the Healthlines intervention is intended to address this limitation. Greenhalgh et al[69] have taken a more radical stance and argued against the quasi-experimental approach advocated by previous authors in favour of in-depth case studies, viewing programme evaluation not as an experimentation but as social practice. They claim that there is a need to recognise the complex political dynamics and language games practiced by different stakeholders and to question rationalist assumptions about 'what works'.[69] We recognise the importance of these political considerations in how telehealth programmes are implemented and evaluated, and in how the findings from such evaluations are sometimes interpreted to fulfil a prior agenda. However, this does not undermine the need to develop interventions based on an understanding of how and in what ways telehealth programmes might be effective; indeed, a clear theoretical basis for interventions and clarity about intended outcomes might provide the most robust defence against selective use of findings and may allow a more nuanced understanding about why interventions are more or less effective in different contexts.

### Implications for clinicians and policymakers

This paper describes a clear conceptual model, based on several sources of evidence, which helps to articulate the theoretical basis for how, why and under what circumstances telehealth could provide specified benefits for patients with chronic health conditions. As it is based on evidence-based components and the views of stakeholders, the TECH model provides the basis for the design of telehealth interventions which are likely to be effective, cost-effective, and acceptable to patients and healthcare providers. Importantly, it also provides a framework for evaluation of these interventions.

**Author affiliations**
[1]University of Bristol, Centre for Academic Primary Care, School of Social and Community Medicine, Bristol, UK
[2]University of Sheffield, Medical Care Research Unit, School of Health and Related Research (ScHARR), Sheffield, UK
[3]University of Southampton, School of Health Sciences, Southampton, UK
[4]University of Southampton, Centre for Applications of Health Psychology, Southampton, UK
[5]Department of General Practice, HRB Centre for Primary Care Research, Royal College of Surgeons in Ireland, Medical School, Dublin 2, Ireland
[6]Division of Psychiatry, University College London, London, UK
[7]NHS Direct, Hampshire, UK
[8]The University of Manchester, Centre for Primary Care, Institute of Population Health, Manchester, UK
[9]Nottingham Clinical Trials Unit, University of Nottingham, Nottingham Health Science Partners, Nottingham, UK

**Acknowledgements** The authors would like to thank the members of the Program Steering Committee, chaired by Professor Brian McKinstry, patient representatives, members of the programme advisory group and other stakeholders who contributed to the different components of research which led to the model described in this paper. They would also like to thank Prof Hayden Bosworth for allowing us to adapt the Duke self-management system and NHS Direct for implementing the intervention which is the basis for the trial.

**Contributors** CS, AO, AR, CP, LY, SH, TF, GL, SL, SB and AAM conceived the idea, developed the protocol, obtained funding and supervised the research. CT and LE managed the research programme. AR, CP and AO conducted the metareview and evidence synthesis. AR and JS conducted the qualitative study. LE conducted the patient survey. All authors contributed to model development. CS drafted the paper which was critically reviewed by all authors. CS is the guarantor.

**Funding** This article presents independent research funded by the National Institute for Health Research (NIHR) under its Programme Grant for Applied Research (grant reference RP-PG-0108-10011).

**Competing interests** None.

**Ethics approval** Southmead Research Ethics Committee.

**Provenance and peer review** Not commissioned; externally peer reviewed.

**Data sharing statement** No additional data are available.

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
