## [Reviewer comments · BMJ Open]

ARTICLE DETAILS

TITLE (PROVISIONAL)	Telehealth in CHronic disease: mixed-methods study to develop the TECH conceptual model for intervention design and evaluation
AUTHORS	Salisbury, Chris; Thomas, Clare; O'Cathain, Alicia; Rogers, Anne; Pope, Catherine; Yardley, Lucy; Hollinghurst, Sandra; Fahey, Tom; Lewis, Glyn; Large, Shirley; Edwards, Louisa; Rowsell, Alison; Segar, Julia; Brownsell, Simon; Montgomery, Alan

VERSION 1 - REVIEW

REVIEWER	Martin Cartwright City University London UK
REVIEW RETURNED	30-Sep-2014

GENERAL COMMENTS	(4.) The paper present a 'conceptual model' that was developed based on several existing (i.e. previously published) studies and two new elements, (i.) a review of existing models of chronic condition management, and (ii.) a review of national guidelines for the management of CVD. To be replicable three separate methods need to be fully articulated: (1.) the methods for the review of the models of chronic conditions, (2.) the methods for the review of national guideline for the management of CVD, and (3.) the methods of synthesising the (quantitative and qualitative) evidence from all the existing studies and from the two additional reviews. Such methods were not articulated clear in the paper. In part it appears that the two new reviews were not systemmatic (and to be fair don't pretend to be systemmatic). In part it is because the process of synthesising all of the evidence was done in a qualitative manner along the lines of a realistic synthesis. This kind of synthesis may never achieve the standards of replicability for other types of review but the authors could offer more transparency in regard to decisons about which elements of evidence were allowed to influence the conceptual model an which were not, and about how the evidence was interpeted and synthesised and how the model emerged. (6.) The objective was to develop a 'conceptual model' for the effective use of telehealth. The term 'conceptual model' is not defined but in several places the authors write as though they are presenting what might more generally be considered an explanatory model (i.e. one that unpacks all of the constructs in the model, explains how the constructs causally relate to each other and how one would change a construct in one part of the model to influence constructs elsewhere in the model). If this is the authors' standpoint then they have overstated their claim - the evidence base for the causal links in this model are, in some cases, highly debatably (see Box 3). For example, the authors claim that all the components in Box 3 have established 'effectiveness' but not necessarily in the
---

context of telehealth, and the authors do not specify what outcome(s) that each of the components is effective at changing. In some cases the components themselves are underspecified - for example, "provide patient information" tells us nothing about the content of the patient information: the content of the patient information needs to be specified in terms of specific behaviour change techniques otherwise it fails to deliver on the authors' claim that their model will help others to design telehealth services. Similar lack of specificity is evident in relation to many of the other components listed in Box 3. The lack of specificity and lack of evidence for many elements of the model means that the model does not really deliver on the authors' promise of

(12.) The strengths and limitations section goes some way to recognising some of the inherent weaknesses but it does not recognise the lack of specificity in terms of the behaviour change techniques that would be required to increase 'engagement', promote self-management etc. in order to optimise the telehealth system (even if we accept that the constructs posited are appropriate - and I would concede that they probably are appropriate, albeit there is a lack of evidence to support this in the context of telehealth).

The model is described as a model for effective use of telehealth in managing chronic conditions (TECH). However, it describes a comprehensive telehealth system that is fully integrated with the wider healthcare system so it is probably a conceptual framework for optimised delivery of telehealth at scale (TECHAS ?) since smaller scale trials of telehealth can be conducted without needing the degree of integration described. It may be useful to make this distinction explicit.

Box 2 raised some questions. For example, what process / criteria were used to classify components/ features of a telehealth system as pre-disposing, enabling or reinforcing? For example, why is 'clarity of roles for conventional and telehealth providers and good communication between them' (p.12) considered a reinforcing factor rather than a pre-disposing factor? The process of arriving at pre-disposing, enabling or reinforcing could be made more transparent. The enabling factor 'patients having a clear understanding of why they have been included' is only relevant to telehealth delivered in the context of a study, it is not relevant to telehealth delivered as part of routine care. It might be helpful to state whether particular components of the model are relevant to telehealth in trials, telehealth in routine care or both - patients' and professionals' barriers and drivers (or pre-disposing, enabling and reinforcing factors) may be different for each context .

The diagrammatic version of the model in Figure 1 reveals a lack of due consideration about the relationship they are trying to show. The model suggest that (promoting patient engagement, promoting provider engagement, promoting self-management, optimising treatment, co-ordinating care, certain patient characteristics & certain features of the wider social and health system) influence chronic disease management [as indicated by the arrows] and in turn chronic disease management influences partnerships between telehealth providers and other healthcare providers and in turn, these partnerships lead to improved outcomes. Clearly this is not what the authors are trying to say but the model is unclear.

	My overall impression is that this model is probably best described as a 'conceptual framework' since it posits some things to consider when attempting to trial telehealth or deliver telehealth in routine care. However, it wants to be more than this, it wants to be a fully articulated explanatory model that specifies causal relationships and informs readers how to intervene to behaviour change (of patients and professionals) in a complex intervention. The model as described is not sufficiently specified to be an explanatory model since it does not explain in a meaningful, evidence-based way how to:  increase patient engagement increase provider engagement increase patient self-management optimise treatment achieve better co-ordination of care accommodate patient characteristics (related to poor outcomes) accommodate features of the wider social and health system (related to poor outcomes) optimise partnerships between telehealth providers and other healthcare providers Until there are precisely specified interventions to address each of these components then this is really a shopping list of ideal features of a telehealth system. The model lacks specificity in places, and may be accused of overstating the evidence based on which it is based (although the authors do say that the strength of the evidence is varied). The model aspires to be an explanatory model but it is probably best described as a conceptual framework (identifying important constructs without specifying the nature of these constructs, demonstrating causal relationships between them or specifying how to intervene to achieve change (increase/ decrease) in particular constructs). As a conceptual framework this could still make a useful contribution to the literature but the authors make claims for the validity and the usefulness of the model beyond what would be acceptable in or areas of the behaviour change literature. Presented the model as a looser 'conceptual framework' would therefore be more defensible at this stage.
--	---

REVIEWER	Hilary Pinnock The University of Edinburgh, UK I am working in a similar area, having completed a number of trials and qualitative work on telehealthcare
REVIEW RETURNED	11-Nov-2014

GENERAL COMMENTS	This paper synthesises the findings of a systematic review, a qualitative study, a patient survey as well as existing models of LTC care and international guidelines to develop a conceptual model for telehealth in the management of people with LTCs. This is a carefully written paper, which has obviously had a lot of thought given to the issue, and is underpinned by the considerable experience of the authors. I have no criticism of the process which is meticulous and robust. Each stage is well described and appropriate. The end-of-project workshop is an excellent strategy for reality checking this type of research.
---

	 • I have been considering the TECH model and my only concern is that I am not sure that it only describes telehealth. The pivotal 'arrow' is the multi-coloured partnership between primary care provider, other healthcare providers and effective telehealth which, the model suggests, could lead to improved outcomes. In fact, all the purple components which feed into this partnership could work without telehealth (and indeed did in our TeleScot COPD telemonitoring trial (BMJ 2013; 347:f6070). This is an extended version of the chronic care model into which telehealth has to fit if it is to become a partner in providing care. Maybe this is what is intended, but perhaps the title of the diagram needs to reflect this. • Should facilitation of self-management be an outcome? This conceptual model now needs to be tested, and the outcome of on-going work will be interesting.
--	--

VERSION 1 – AUTHOR RESPONSE

Reviewer 1

(4) The main empirical work and focus of this research programme was the first three sources of evidence we describe: the meta-review, patient survey and qualitative research. However, in developing our conceptual model we also wanted to (a) take account of existing models of chronic disease management (CDM) and (b) apply our telehealth model to recognised priorities for intervention for our two exemplar conditions. We are not suggesting that our reviews of CDM models or of international guidelines were akin to systematic reviews. We have revised this section of the method (page 8) to describe more clearly the process that we used to make it clearer that these were part of the process of developing the model and applying it to our exemplar conditions, rather than major pieces of empirical work. We also needed to scope existing models of chronic disease management in order to discuss the strengths and weaknesses of our new approach. We have now referenced the relevant disease management guidelines from the UK, US and Europe so that readers can replicate our findings (page 10).

The reviewer has requested more information about the methods we used to synthesise the findings from the various strands of evidence. We think we have described the process clearly in the section headed 'synthesis'. This was an iterative approach. We reported the findings from the various studies and discussed them within the research team, seeking to show hypothesised relationships between different various constructs in a schematic manner. This process evolved over several iterations, as we discussed, critiqued and sought to improve the model. Finally, we discussed it with a wide range of stakeholders in a workshop involving small group discussions and feedback, and used this to develop the model further to create the final version. We note that the second reviewer had a very different view from the first reviewer, commenting that our process for developing the model was 'meticulous and robust', 'well described and appropriate' and 'an excellent strategy for developing this type of model'. However, we have slightly expanded this section of the paper in an effort to make the process clearer (pages 8-9).

(6) We think that reviewer one has a particular understanding of the word 'model', and is thinking of tightly specified psychological models of behaviour change. However the word 'model' is used in a wide range of ways in different fields. We have not claimed that ours is a behaviour change model. Instead, it is a model to guide organisational design and service delivery. Our model is comparable with the Chronic Care Model[1], the NHS Long Term conditions model[2], and the House of Care Model.[3] All of these models are widely used (and are presumably therefore useful), even though they are arguably less tightly specified than our TECH model and their development did not appear to follow such an explicit and evidence based process as we have used. We have now added a short paragraph to explain that ours is not a behaviour change model. (see original page 6, and new

paragraph pages 18-19)

Reviewer 1 suggests we use the term 'framework', but we still contend that 'model' is a more appropriate term. Firstly, models are often developed to assist with service design and delivery, with these types of models focussing on what works in a given situation for a given group of people. This is just the sort of function we anticipate our model serves. In addition, we do think it is possible to hypothesise causal relationships between the components in our model, such as that patient engagement will lead to better treatment outcomes through greater usage of telehealth. Indeed, as we explain in the paper (page 6, unchanged), one of the main purposes of developing a model was to explore these relationships in our on-going evaluation of the Healthlines intervention which was based on this model. To this end, we have included measures of each of these components in the evaluation of the intervention, and we will explore whether they were delivered to patients as intended and whether this appeared to be associated with the intended outcomes.

We have tried to make it clearer throughout the paper (abstract; page 9; page 12; page 17) that the TECH model describes a series of key attributes or components that we hypothesise (based on our several sources of evidence) are likely to be important in developing telehealth systems that are effective (that is, to lead to outcomes that we specify in the model). Given that we are trying to design a generalisable model which can be applied in different health care systems, with different telehealth interventions, and in different chronic diseases, it would not be possible to specify all of the individual components in detail. Box 3 does provide some specificity by listing strategies within each component which have been shown to be associated with improved patient outcomes in a range of conditions, although we already state in the discussion that the evidence for some of them is variable.

(12) Please see response to (6). We accept that there is a lack of evidence to support some of these constructs in the context of telehealth (as we said on page 18). That is precisely why we have developed a model: in order to make explicit the various constructs that we hypothesise are likely to be important, so that we can test them and provide evidence which supports or refutes the model.

Reviewer one also wonders whether we should describe this as a conceptual framework for 'optimised delivery of telehealth at scale'. We think that phraseology would be inappropriate (because our model is about content as well as service delivery) and unnecessarily cumbersome. We aimed to develop a model for effective use of telehealth in managing chronic conditions, and there is little point in developing this model if it is not to be used at scale in wider health care systems. However we have now made this point explicit in the text (pages 18-19).

The questions raised around Box 2 are important and we thank the reviewer for bringing to our attention the need for further clarification of the PRECEDE PROCEED process. We agree that more detail would be helpful for the reader, and so have added some of the following information to the first paragraph of the 'Synthesis' sub-section of the revised manuscript, as well as more clearly described how the key themes were categorised into the causal factors (page 8).

The PRECEDE-PROCEED model provides specific definitions and detailed explanations of what a predisposing, enabling, and reinforcing factor is. Predisposing factors are those that lead to a specific behaviour, or provide the motivation to act in some way. Enabling factors are those that make it possible to carry out the action. Reinforcing factors act like social feedback mechanisms, in which either positive or negative cues are perceived (e.g., through social reinforcement), and then go on to influence the likelihood that one will perform the behaviour in the future. In terms of the example raised by the reviewer, where roles for conventional and telehealth providers are clearly understood, and there is also good communication between them, this will help to perpetuate engagement with the telehealth service; in other words, it will reinforce this behaviour. If communication were poor, primary care health professionals may feel isolated from patients, unclear of what treatments they are

undergoing, unsure of whether the telehealth service is benefiting patients or not, and so probably disengage with it. Similarly, having clearly defined roles enhances job satisfaction and performance, which are sources of positive feedback. While it is important that the roles of health professionals are clearly defined and communicated at the outset of introducing a new service (a predisposing factor to initial engagement with it), the subsequent satisfaction and performance-enhancing effects of having such clear roles is what we were focussing on when classifying this as a reinforcing factor. As demonstrated by this example, the same information can serve as a predisposing factor initially, and then later as a reinforcing factor. The real importance of classifying information into these types of causal factors is to devise temporally-appropriate strategies to enhance motivators of and mitigate barriers to the target behaviour.

The second example raised by the reviewer was the pre-disposing factor 'patients having a clear understanding of why they have been included'. We suggest that the extent that a patient understands why they are being offered treatment, what the treatment involves, potential benefits and drawbacks, and so on will necessarily influence their choice to take part in the treatment, adherence to the treatment, and potential benefit gained from it. This is equally likely to be true in a routine treatment as in a study. We acknowledge that the wording we had used in Box 2 could have been interpreted to be study-specific, so we have altered this to make it clear that this is applicable to any sort of telehealth treatment. This now reads as, 'patients having a clear understanding of why they have been offered telehealth treatment' (page 12)

Figure 1: The reviewer's comments are helpful in pointing out that the nature of some of the relationships shown using arrow in Figure 1 may not be sufficiently clear. There is always a difficulty in designing diagrammatic models which are simple and memorable but also sufficiently detailed and clear. Figure 1 was intended to show that interventions to promote self-management, optimisation of treatment and care co-ordination are all essential aspects of chronic disease management, which are likely to lead to improved health outcomes, patient experience, access to care and more cost-effective delivery of care. These benefits are more likely to be achieved if the service is delivered in an integrated way with other health care providers, and the effectiveness of telehealth is likely to be moderated by the extent of patient and provider engagement and also moderated by characteristics of patients and the health care system. We have changed the layout of Figure 1 and the text of the results section (page 12) to make this clearer.

Reviewer 2.

In contrast to Reviewer 1, this reviewer was very positive about our paper and felt that the paper was carefully written and thoughtful. She felt that the research was well conducted and well described.

We agree that the components of our model have relevance extending beyond telehealth alone. We have now added this point to the text (page 18). However, our evidence review shows that most telehealth interventions are designed without giving attention to many of these components (e.g. the need for partnership with existing primary care providers, or the importance of patient engagement) and we suggest in the paper that might be why the outcomes from previous telehealth interventions have often been disappointing. Our model is intended to draw attention to components which appear to be essential in telehealth, and this can be used to design and evaluate telehealth interventions.

We thank the reviewer for the suggestion that facilitation of self-management should be an outcome. We are sympathetic to this idea, but it seems rather ad hoc to add this to Figure 1 since it did not arise from the process of development of the model that we have been careful to describe. Furthermore it could be argued that self-management is an intermediate outcome rather than an end in itself, in that it should lead to improved health outcomes and/or more cost-effective care.

1. Wagner EH, Austin BT, Von KM. Improving outcomes in chronic illness. *ManagCare Q* 1996;4(2):12-25.
2. The NHS and Social Care long term conditions model. Secondary The NHS and Social Care long term conditions model.
http://webarchive.nationalarchives.gov.uk/+www.dh.gov.uk/en/Healthcare/Longtermconditions/DH_4130652.
3. Coulter A, Roberts S, Dixon A. Delivering better services for people with long-term conditions: Building the house of care. Secondary Delivering better services for people with long-term conditions: Building the house of care 2013.
http://www.kingsfund.org.uk/sites/files/kf/field/field_publication_file/delivering-better-services-for-people-with-long-term-conditions.pdf.

VERSION 2 – REVIEW

REVIEWER	Hilary Pinnock The University of Edinburgh, Scotland
REVIEW RETURNED	12-Dec-2014

GENERAL COMMENTS	Thank you for responding to my comments. I believe this is a useful contribution to thinking about telehealthcare and its integration into care of people with long-term conditions. I think it will be useful not only for the design of interventions, but also for explaining why interventions are (or are not) effective
---